# Retrieval-Augmented Few-shot Text Classification

**Guoxin Yu**🍓🍇🍋*, **Lemao Liu**🍊†, **Haiyun Jiang**🍊, **Shuming Shi**🍊, **Xiang Ao**🍓🍋🍒†

🍓Key Lab of Intelligent Information Processing of Chinese Academy of Sciences (CAS),
Institute of Computing Technology, CAS, Beijing 100190, China.
🍇Peng Cheng Laboratory.
🍋University of Chinese Academy of Sciences, Beijing 100049, China.
🍊Tencent AI Lab, China. 🍒Institute of Intelligent Computing Technology, Suzhou, CAS.
{yuguoxin20g,aoxiang}@ict.ac.cn
{redmondliu, haiyunjiang, shumingshi}@tencent.com

## Abstract

Retrieval-augmented methods are successful in the standard scenario where the retrieval space is sufficient; whereas in the few-shot scenario with limited retrieval space, this paper shows it is non-trivial to put them into practice. First, it is impossible to retrieve semantically similar examples by using an off-the-shelf metric and it is crucial to learn a task-specific retrieval metric; Second, our preliminary experiments demonstrate that it is difficult to optimize a plausible metric by minimizing the standard cross-entropy loss. The in-depth analyses quantitatively show minimizing cross-entropy loss suffers from the weak supervision signals and the severe gradient vanishing issue during the optimization. To address these issues, we introduce two novel training objectives, namely EM-L and R-L, which provide more task-specific guidance to the retrieval metric by the EM algorithm and a ranking-based loss, respectively. Extensive experiments on 10 datasets prove the superiority of the proposed retrieval augmented methods on the performance.

## 1 Introduction

Few-shot text classification, which entails learning a new task based on limited training data, has been advanced by pre-trained language models (PLMs) (Brown et al., 2020; Liu et al., 2023) and prompt engineering (Gao et al., 2021; Chen et al., 2022a). However, since training numerous parameters of PLMs on scarce data is prone to produce over-fitting (Liu et al., 2021) and unstable generalization, only using the trained parameters for inference usually leads to unsatisfactory performance on unseen test data.

On the other hand, retrieval-based methods have witnessed success on various natural language processing tasks, thanks to their capability of incorporating *retrieved memory* alongside parameters for better generalization. These methods retrieve relevant examples as memories from a large-scale corpus through either a static retrieval metric (Lewis et al., 2020; Wang et al., 2022) or a joint learning-based metric (Cai et al., 2021; Siriwardhana et al., 2023) and then the retrieved examples are used to make a prediction. In this way, their generalization ability is achieved by not only the model parameters but also the retrieved memory.

Despite the theoretical potential of promoting generalization by using retrieved memory, previous retrieval-augmented methods empirically struggle to showcase compelling ability in few-shot learning scenarios, where the retrieval space (i.e., the few-shot training data) is limited. Specifically, static retrieval may lack neighbors with high metrics in the case of limited retrieval space. Even though such neighbors exist, static retrieval cannot be reliable for retrieving really helpful samples for target prediction, because its metric is not task-specific. In particular, for joint learning-based retrieval which minimizes the standard cross-entropy based loss, although the retrieval metric is updated towards the downstream task, it suffers from the gradient vanishing problem during the optimization process as quantitatively measured in Fig. 2 (see §5.2 later). As a result, in a few-shot scenario, the retrieval metric might be not optimized well due to insufficient training data.

To overcome the aforementioned challenges, we propose two novel training objectives, namely Expectation Maximization-based Loss (EM-L) and Ranking-based Loss (R-L), for learning to retrieve

---

*Work done while this author was an intern at Tencent.
†Corresponding authors.

examples from a limited space more effectively. Both objectives are committed to obviating the gradient vanishing problem and prioritizing more beneficial examples for specific downstream tasks. In the EM-L approach, the retrieved examples are treated as latent variables, and an iterative process of Expectation-step and Maximization-step is employed until convergence (Dempster et al., 1977). The posterior distribution of the latent variable is estimated to measure the importance of candidate examples in the E-step, while the M-step maximizes the expectation log-likelihood. By approximating the retrieval metric according to the posterior probability, more productive examples could be recalled for downstream tasks with limited training data.

Following a similar idea, R-L optimizes an additional ranking loss function to provide more direct supervision to the examples retriever, which draws inspiration from pair-wise ranking algorithm (Freund and Schapire, 1997; Burges et al., 2005; Rudin and Schapire, 2009). Such a tailored loss measures the consistency between the retrieval metric and the auxiliary function associated with each example for classification purposes. Minimizing the loss could effectively strengthen the supervision signals for the example retriever.

Our experimental evaluation on ten text classification datasets demonstrates the superiority of EM-L and R-L over existing retrieval methods within a limited retrieval space. The comparative analyses further confirm that EM-L and R-L alleviate the weak supervisory signals and gradient vanishing issue suffered by joint learning-based retrieval. Our contributions could be summarized as follows:

- We discuss the weak supervision signals and gradient vanishing problem encountered by existing retrieval methods minimizing the standard cross-entropy loss, as quantitatively measured in §5.2.

- We introduce two novel training objectives, namely EM-L and R-L, which optimize the retriever more effectively, thus recalling more productive examples from a limited space.

- Extensive experiments and analyses demonstrate that the proposed methods achieve better performance on few-shot text classification and alleviate the supervision insufficiency and gradient vanishing issues.

## 2 Revisiting Retrieval-augmented Methods in Few-shot Learning

### 2.1 Retrieval-augmented Methods

In this paper, we revisit the retrieval-augmented methods in few-shot text classification and formulate the task in a general framework. Our primary objective is to retrieve examples from limited training data to improve the few-shot text classification.

**Model Formulation** All retrieval methods could comprise an example retriever and a text classifier. We provide the formal formulation inspired by Singh et al. (2021) and Izacard et al. (2022):

$$
\begin{aligned}
P_{\theta,\phi}(y|\mathbf{x}) &= \sum_{j=1}^{m} P_\theta(y|\mathbf{x}, \mathbf{z}_j) P_\phi(\mathbf{z}_j|\mathbf{x}), \\
P_\theta(y|\mathbf{x}, \mathbf{z}_j) &= \mathsf{softmax}(f_{\mathrm{clf}}(\mathbf{x} \oplus \mathbf{z}_j)), \\
P_\phi(\mathbf{z}_j|\mathbf{x}) &= f_{\mathrm{retr}}(\mathbf{x}, \mathbf{z}_j),
\end{aligned}
\tag{1}
$$

where $\mathbf{x}$ and $\mathbf{z}_j$ denote the representations of original input and a retrieved example from the training set, and $y$ corresponds to the class associated with input $x$. $f_{\mathrm{clf}}$ and $f_{\mathrm{retr}}$ serve as the text classifier and the example retriever, which selects examples according to a retrieval metric. $\theta$ and $\phi$ denote the trainable parameters of the text classifier and examples retriever. $m$ is a hyperparameter that denotes the number of fetched examples. The operation $\oplus$ signifies concatenation, and the term $\mathsf{softmax}$ refers to the normalized exponential function. Specifically, $\mathbf{z}$ corresponds to a set of retrieval examples, which can either be $\{\langle x^s, y^s \rangle\}$ pairs or $\{x^s\}$. The latter form is adopted in this paper for simple experiments.

The standard cross entropy is employed to optimize the classifier and example retriever as follows:

$$
\mathcal{L} = -\sum_{i=1}^{n} \log P_{\theta,\phi}(y_i|\mathbf{x}_i), \tag{2}
$$

where $n$ is the total number of training instances and $y_i$ is the gold label of the $i$-th instance. During inference, for all retrieval methods, we select top $m$ examples according to $P_\phi(\mathbf{z}_j|\mathbf{x})$ and get the final classification results using the first line of Eq. (1).

**Static Retrieval** Given an input sentence $\mathbf{x}$ and a retrieval corpus, static retrieval aims to search for a set of relevant examples $\mathbf{Z}$ according to a fixed retrieval metric (Borgeaud et al., 2022; Wang et al., 2022; Li et al., 2022). Following the Eq. (1), its

retrieval metric is defined as follows:

$$P_\phi(\mathbf{z}_j|\mathbf{x}) = f_{\text{retr}}(\mathbf{x}, \mathbf{z}_j) = \text{sim}(\mathbf{x}, \mathbf{z}_j). \quad (3)$$

Here, $\text{sim}(\mathbf{x}, \mathbf{z}_j)$ represents a fixed metric without any trainable parameters, such as TF-IDF (Sparck Jones, 1972), BM25 (Robertson et al., 2009), and semantic similarity encoded by PLMs. Such fixed metrics cannot adapt to the downstream task and prioritize the most helpful examples. Particularly, this limitation will be amplified in few-shot learning with scarce training data.

**Joint Learning based Retrieval** Static retrieval assumes that higher similarity between $\mathbf{z}_j$ and $\mathbf{x}$ implies a greater auxiliary effect of $\mathbf{z}_j$ on $\mathbf{x}$. However, the assumption failed to hold in tasks where inputs with high similarity have distinct labels, such as sentiment classification. To address this limitation, joint learning-based retrieval (Cai et al., 2021; Gao et al., 2022; Siriwardhana et al., 2023) unifies the retriever and the downstream model to jointly train them for specific tasks. Following Eq. (1),

$$P_\phi(\mathbf{z}_j|\mathbf{x}) = f_{\text{retr}}(\mathbf{x}, \mathbf{z}_j) = \frac{\exp(\mathbf{x} \cdot \mathbf{z}_j^\top)}{\sum_{j=1}^m \exp(\mathbf{x} \cdot \mathbf{z}_j^\top)}. \quad (4)$$

$f_{\text{retr}}(\mathbf{x}, \mathbf{z}_j)$ is a trainable dot product attention. Notably, the absence of ground truth for $P_\phi(\mathbf{z}_j|\mathbf{x})$ makes it challenging to determine which $\mathbf{z}_j$ is the most beneficial one, and it relies implicitly on distant supervision from text classification.

Both static retrieval and joint learning-based retrieval are proposed to retrieve examples from a large-scale corpus. In this paper, we mainly focus on few-shot text classification and retrieve the most helpful examples from the limited training set.

## 2.2 Challenges in Few-shot Learning

While the above retrieval-augmented methods have shown advancements in various natural language processing tasks, their performance in few-shot learning remains unconvincing. In other words, retrieving examples from a narrow space to improve few-shot learning is still challenging due to limited training data. Previous studies (Li et al., 2022; Siriwardhana et al., 2023) have revealed that static retrieval may not fetch the most helpful examples in tasks where similar inputs correspond to different labels, primarily due to their unreasonable assumption that higher similarity implies better suitability for the downstream task. Moreover, we also find

static retrieval even underperforms methods without retrieval in some few-shot tasks (see Table 1). Such failure can also be attributed to data limitation in few-shot scenarios, where examples with high static similarities are scarce or non-existent.

In addition, joint learning-based retrieval methods (Ren et al., 2021; Cai et al., 2021; Siriwardhana et al., 2023) are good solutions to enhance the adaptability of the retrieval to downstream tasks. However, our study demonstrates that learnable metrics struggle to be trained as anticipated and are inferior to static metrics in several few-shot tasks (see Table 1). The main underlying factors are the scarcity of data and the weak supervision signals provided to the learnable retrieval metric. In more detail, the retrieval metrics in joint learning-based methods are adjusted solely based on distant supervision from the downstream tasks, which is significantly further weakened by the limited data. This fact is further supported by quantifying the gradient of retrieval parameters: the gradient norm of the parameters in retrieval metric is more than $1e-6$ for only about $40\%$ updates in some datasets as shown in Figure 2 (see §5.2 later).

In this paper, our objective is to meet the challenges of weak supervision signals for the retriever and insufficient data, aiming to retrieve the most helpful examples to promote model generalization.

## 3 Methodology

### 3.1 Overview

Given the limitations posed by limited data and weak supervision signals, existing retrieval methods are inadequate for addressing these challenges. To address these limitations, we propose two novel training objectives, which are achieved by two loss functions: Expectation Maximization-based Loss (EM-L) and Ranking-based Loss (R-L). Both methods aim to enhance the retrieval quality by giving the retriever more supervisory signals and prioritizing examples that are more beneficial for the specific task with limited training data. In essence, we seek to maximize the consistency between the metric distribution $P(\mathbf{z}_j|\mathbf{x})$ and the classification distribution $P(y|\mathbf{x}, \mathbf{z}_j)[y_i]$ as much as possible. In this way, more suitable examples are retrieved and the performance of text classification could be improved even in the few-shot scenario. Additionally, we integrated EM-L, R-L, and two existing retrieval methods with two popular text classification backbones to compare their respective performance.

## 3.2 Backbone

**Fine-tune Pre-trained Language Models** For each sentence, we use PLMs to tokenize the input sentence into $\{[\text{CLS}], x_1, ..., x_l, [\text{SEP}]\}$ with $(l + 2)$ tokens and extract the representation $\mathbf{x}$ of [CLS] as the sentence embedding. In the same way, the $j$-th retrieved example is represented as $\mathbf{z}_j$. These tensors are subsequently fed into the example retriever and classifier, producing the final probability estimated for label $y$.

**Prompt Learning** Another backbone is to transform the text classification into a cloze question problem (Schick and Schütze, 2021). Let $\mathcal{M}$ be a masked language model with vocabulary $\mathcal{V}$, and $\mathcal{Y}$ denote the label set of a specific downstream task $A$. Prompt learning employs a function $\mathcal{P}$ to convert an input sentence into a phrase containing a prompt with a [MASK] token. Then an injective function $v : \mathcal{L} \rightarrow \mathcal{V}$ is utilized to map each label to a word from $\mathcal{M}$'s vocabulary $\mathcal{V}$. We first obtain the representation of [MASK] and determine the most suitable word from $\mathcal{V}$ for filling the [MASK]. For instance, the application of prompt learning to sentiment classification can be outlined as follows:

$$\mathcal{P}(x) = \{[\text{CLS}], x_1, ..., x_l, \text{it was } [\text{MASK}], [\text{SEP}]\}$$
$$P(y|\mathbf{x}) = g(P([\text{MASK}] = v(y)|\mathbf{x})),$$
$$v(y) \in \{\text{great}, \text{terrible}\},$$
$$\tag{5}$$

where $\mathbf{x}$ is the representation of [MASK], $g$ converts the probability of label words to classes, and $l$ is sentence length. The representation $\mathbf{z}_j$ of a retrieved example is yielded from a [MASK] token in the same way.

## 3.3 Expectation Maximization-based Loss (EM-L)

Considering the absence of the ground truth for $P_\phi(\mathbf{z}_j|\mathbf{x})$ in Eq. (1), we regard $\mathbf{z}$ as a latent variable and propose an EM-based retrieval objective to estimate $P_\phi(\mathbf{z}_j|\mathbf{x})$. This method alternates between an Expectation-step and a Maximization-step until convergence. In the E-step, the current parameters are used to estimate the posterior distribution of the latent variable given the observed data. Specifically, we retrieve $m$ examples from the training set and compute the conditional probabilities of the latent variable using:

$$P_{\theta,\phi}(\mathbf{z}_j|\mathbf{x}, y) = \frac{P_\theta(y|\mathbf{x}, \mathbf{z}_j)P_\phi(\mathbf{z}_j|\mathbf{x})}{\sum_{j=1}^m P_\theta(y|\mathbf{x}, \mathbf{z}_j)P_\phi(\mathbf{z}_j|\mathbf{x})},$$
$$\tag{6}$$

where $P_\theta(y|\mathbf{x}, \mathbf{z}_j)$ and $P_\phi(\mathbf{z}_j|\mathbf{x})$ are obtained from classifier $f_{\text{clf}}$ and examples retriever $f_{\text{retr}}$ in Eq. (1) respectively. $m$ denotes the number of retrieved examples.

In the M-step, the parameters are updated by maximizing the expected log-likelihood, which is taken with respect to the estimated posterior $P_{\theta,\phi}(\mathbf{z}_j|\mathbf{x}, y)$ in the E-step:

$$P_{\theta,\phi}(y|\mathbf{x}) = \sum_{j=1}^m P_{\theta,\phi}(\mathbf{z}_j|\mathbf{x}, y) \cdot \log P_\theta(y|\mathbf{x}, \mathbf{z}_j).$$
$$\tag{7}$$

Since we sample $m$ examples from the training set by $P_\phi(\mathbf{z}_j|\mathbf{x})$ and estimate $P_{\theta,\phi}(\mathbf{z}_j|\mathbf{x}, y)$ based on $m$ examples in the E-step, more supervision will be provided to the retriever during the optimization in the M-step. Please refer to Appendix A for proof of rationality of Eq.(6) and why EM-L can minimize the likelihood-based loss defined in Eq. (2).

## 3.4 Ranking-based Loss (R-L)

Following the main idea claimed in § 3.1, Ranking-based Loss (R-L) considers the process of retrieving $\mathbf{z}_j$ as a ranking task. Unlike EM-L, R-L employs a ranking loss to enhance the consistency between $P_\theta(y|\mathbf{x}, \mathbf{z}_j)[y_i]$ and $P_\phi(\mathbf{z}_j|\mathbf{x})$ and provide more direct signals to the retriever. The optimization objective of R-L aims to ensure that $\mathbf{z}_j$ with higher $P_\theta(y|\mathbf{x}, \mathbf{z}_j)[y_i]$ has higher $P_\phi(\mathbf{z}_j|\mathbf{x})$ by minimizing the following $\mathcal{L}_R$:

$$\mathcal{L}_R = \sum_i^n \sum_j^m \max(P_\theta(y|\mathbf{x}_i, \mathbf{z}_j)[y_i]$$
$$- P_\phi(\mathbf{z}_j|\mathbf{x}_i) + \delta, 0).$$
$$\tag{8}$$

Here, $P_\theta(y|\mathbf{x}, \mathbf{z}_j)$ and $P_\phi(\mathbf{z}_j|\mathbf{x})$ are obtained from $f_{\text{clf}}$ and $f_{\text{retr}}$ in Eq. (1), $m$ and $n$ denote the number of retrieved examples and training instances. $\delta$ is a margin parameter imposing the distance between two distributions to be larger than $\delta$.

The ranking loss $\mathcal{L}_R$ is added to the overall loss $\mathcal{L}$ in Eq. (2) with a weight $\lambda$ every $t$ step:

$$\mathcal{L}_{sum} = \mathcal{L} + \lambda \cdot \mathcal{L}_R,$$
$$\lambda = \begin{cases} 1, & \text{step} \mod t = 0; \\ 0, & \text{otherwise}; \end{cases}$$
$$\tag{9}$$

where $\lambda > 0$ is a hyperparameter to trade off both loss terms, and step denotes the training steps.

| Model | Single Sentence | | | | Sentence Pair | | | | ABSA | | Avg. |
|---|---|---|---|---|---|---|---|---|---|---|---|
| | SST-2 | MR | CR | TREC | QQP | QNLI | MNLI | SNLI | RES | LAP | |
| *Prompt Learning with RoBerta-Large* | | | | | | | | | | | |
| Vanilla | $84.84_{(6.80)}$ | $77.88_{(7.90)}$ | $88.36_{(2.89)}$ | $87.20_{(7.70)}$ | $67.09_{(6.70)}$ | $64.25_{(7.45)}$ | $60.69_{(4.08)}$ | $64.56_{(4.08)}$ | $72.05_{(4.08)}$ | $71.81_{(2.88)}$ | $73.87$ |
| Static | $88.60_{(4.10)}$ | $83.67_{(6.80)}$ | $87.06_{(3.84)}$ | $90.95_{(1.36)}$ | $68.31_{(7.70)}$ | $66.27_{(4.98)}$ | $60.38_{(6.70)}$ | $68.17_{(5.62)}$ | $70.95_{(5.46)}$ | $73.01_{(3.03)}$ | $75.74$ |
| Joint | $90.71_{(1.20)}$ | $85.83_{(2.40)}$ | $86.76_{(6.50)}$ | $90.57_{(4.17)}$ | $67.26_{(4.40)}$ | $63.15_{(7.16)}$ | $61.95_{(4.65)}$ | $67.64_{(5.80)}$ | $71.07_{(2.97)}$ | $73.32_{(2.26)}$ | $75.83$ |
| EM-L | $\underline{91.31}_{(1.30)}$ | $\underline{87.58}_{(1.40)}$ | $\mathbf{90.00}_{(0.90)}$ | $\underline{92.13}_{(1.41)}$ | $\mathbf{74.41}_{(0.74)}$ | $\mathbf{67.66}_{(3.77)}$ | $\underline{64.85}_{(3.21)}$ | $\underline{69.52}_{(3.69)}$ | $\underline{73.74}_{(3.46)}$ | $\mathbf{76.02}_{(1.90)}$ | $\underline{78.72}$ |
| R-L | $\mathbf{91.58}_{(1.30)}$ | $\mathbf{87.47}_{(0.09)}$ | $\underline{89.93}_{(1.70)}$ | $\mathbf{92.86}_{(1.21)}$ | $\underline{73.79}_{(2.28)}$ | $\underline{67.62}_{(5.79)}$ | $\mathbf{66.04}_{(3.18)}$ | $\mathbf{73.08}_{(4.59)}$ | $\mathbf{76.79}_{(2.60)}$ | $\underline{75.59}_{(1.51)}$ | $\mathbf{79.46}$ |
| *Fine-tune RoBerta-Large* | | | | | | | | | | | |
| Vanilla | $81.59_{(4.50)}$ | $73.59_{(9.90)}$ | $81.63_{(4.08)}$ | $85.95_{(5.57)}$ | $61.42_{(8.19)}$ | $57.20_{(2.09)}$ | $59.90_{(5.72)}$ | $59.19_{(5.58)}$ | $69.21_{(4.14)}$ | $71.06_{(5.11)}$ | $70.07$ |
| Static | $81.99_{(10.8)}$ | $72.69_{(5.05)}$ | $82.75_{(5.50)}$ | $87.02_{(3.25)}$ | $60.23_{(9.60)}$ | $57.11_{(3.90)}$ | $54.69_{(4.78)}$ | $62.65_{(5.10)}$ | $70.48_{(8.74)}$ | $71.37_{(3.03)}$ | $70.10$ |
| Joint | $83.49_{(3.20)}$ | $74.89_{(2.90)}$ | $80.63_{(5.42)}$ | $86.33_{(3.17)}$ | $63.50_{(8.08)}$ | $57.66_{(2.69)}$ | $60.99_{(4.98)}$ | $61.01_{(5.80)}$ | $70.23_{(3.57)}$ | $70.62_{(4.47)}$ | $70.94$ |
| EM-L | $\mathbf{85.38}_{(1.30)}$ | $\mathbf{75.80}_{(2.20)}$ | $\mathbf{83.81}_{(5.36)}$ | $\mathbf{89.36}_{(2.64)}$ | $\underline{65.70}_{(8.17)}$ | $\underline{60.93}_{(1.56)}$ | $\mathbf{62.24}_{(3.12)}$ | $\underline{65.25}_{(3.20)}$ | $\underline{71.64}_{(3.36)}$ | $\mathbf{72.69}_{(3.18)}$ | $\underline{73.27}$ |
| R-L | $\underline{84.69}_{(2.29)}$ | $\underline{75.35}_{(2.20)}$ | $\underline{83.17}_{(3.22)}$ | $\underline{88.92}_{(3.81)}$ | $\mathbf{70.53}_{(2.68)}$ | $\mathbf{61.37}_{(0.12)}$ | $\underline{62.18}_{(1.72)}$ | $\mathbf{66.31}_{(3.30)}$ | $\mathbf{73.28}_{(3.13)}$ | $\mathbf{72.69}_{(3.01)}$ | $\mathbf{73.85}$ |

Table 1: Comparison results on 16-*shot* text classification. "Vanilla" denotes methods without retrieval, which only consists of a sentence encoder and a classifier. "Static" and "Joint" are static retrieval and joint learning-based retrieval, which are introduced in §2. "EM-L" and "R-L" are methods implemented with our proposed new objectives. All the reported results are average *Accuracy* and the standard deviation in the subscript.

## 4 Experimental Results

### 4.1 Experimental Settings

**Datasets** We compared the proposed EM-L and R-L approaches with existing retrieval methods by conducting experiments on 10 widely used text classification datasets, including single-sentence classification, sentence pair classification, and aspect-based sentiment classification. We created few-shot datasets following Gao et al. (2021). For more details, please refer to Appendix B.

**Baselines** To prove the effectiveness of retrieving examples from the training set, we develop a baseline method without retrieval for comparison. It comprises an input encoder described in § 3.2 and a feed-forward neural network for classification. For comparing different retrieval methods, we evaluated our EM-L and R-L against static retrieval and joint learning-based retrieval. We combine them with two widely used backbones for text classification: pre-trained language models fine-tuning and prompt learning. Please refer to Appendix C for more implementations, such as hyper-parameters and templates in prompt learning.

**Evaluation.** We evaluate all the retrieval methods using two metrics: *Accuracy* and *Kendall's* $\tau$. *Accuracy* represents the proportion of correctly classified instances out of the total number of instances. *Kendall's* $\tau$ is employed to measure the consistency and correlations between the retrieval metric $P_\phi(\mathbf{z}|\mathbf{x}_i)$ and its auxiliary $P_\phi(y|\mathbf{x}_i, \mathbf{z})[y_i]$

for classification. *Kendall's* $\tau$ is defined as follows:

$$\tau_i =$$
$$\frac{2}{m(m-1)} \sum_{j<k}^{m} sign(u_j - u_k) \cdot sign(v_j - v_k),$$
$$u \sim P_\phi(\mathbf{z}|\mathbf{x}_i), v \sim P_\phi(y|\mathbf{x}_i, \mathbf{z})[y_i], \tau_i \in [-1, 1], \quad (10)$$

where $sign(\cdot) \in \{-1, 0, 1\}$ is a sign function. A ranking pair $\langle j, k \rangle$ is concordant if their ranks have the same order in $P_\phi(\mathbf{z}|\mathbf{x}_i)$ and $P_\phi(y|\mathbf{x}_i, \mathbf{z})[y_i]$. Consequently, a positive $\tau_i$ indicates a positive correlation between two distributions, and vice versa. For $n$ instances $\mathbf{x}_i$ in the training set, we calculate the proportion of $\mathbf{x}_i$ with $\tau_i > 0$ as follows:

$$\tau' = \frac{\sum_i^n step(\tau_i)}{n},$$
$$step(\tau_i) = \begin{cases} 0, & \tau_i \leq 0 \\ 1, & \tau_i > 0 \end{cases}. \quad (11)$$

The reported *Kendall's* $\tau'$ in the following experiment is actually $\tau'$, which represents the proportion of instances with $\tau_i > 0$.

### 4.2 Main Results

The experimental results for 16-shot setting on 10 datasets are reported in Table 1, where different retrieval-based methods are combined with two backbones. Several insightful observations could be drawn from the results.

*Retrieving examples from the training set is effective in few-shot scenarios.* Firstly, in most datasets, retrieval-augmented models outperform the vanilla

| Kendall's $\tau'$ | SST-2 | CR | QQP | QNLI | RES |
|---|---|---|---|---|---|
| Static | 0.5344 | 0.5837 | 0.4307 | 0.5312 | 0.47857 |
| Joint | 0.5413 | 0.6129 | 0.4776 | 0.5937 | 0.4732 |
| EM-L | *0.6853* | *0.6451* | **0.6265** | **0.7500** | **0.6598** |
| R-L | **0.7442** | **0.6562** | *0.6057* | *0.7185* | *0.6125* |

Table 2: *Kendall's $\tau'$ of $P_\phi(\mathbf{z}_j|\mathbf{x}_i)$ and $P_\theta(y|\mathbf{x}_i, \mathbf{z}_j)[y_i]$.*

| Accuracy | SST-2 | MR | TREC | QQP |
|---|---|---|---|---|
| Vanilla | 80.22 | 60.71 | 86.05 | 64.27 |
| Static | 76.58 | 67.51 | 86.94 | 60.30 |
| Joint | 85.41 | 71.01 | 86.57 | 61.92 |
| EM-L | *87.30* | **78.75** | *87.52* | **67.90** |
| R-L | **89.79** | *77.38* | **88.78** | *66.77* |

Table 3: Comparison results on 8-*shot* text classification. Standard deviations are omitted to save space.

| | MR | TREC | RES | LAP |
|---|---|---|---|---|
| | *Accuracy* | | | |
| Vanilla | 90.80 | 96.80 | 86.53 | 80.87 |
| Static | 91.40 | 97.60 | 87.50 | 81.19 |
| Joint | 90.90 | 97.80 | 87.58 | 82.13 |
| EM-L | *91.70* | **98.00** | *88.04* | *82.76* |
| R-L | **91.45** | **98.00** | **88.48** | **83.22** |
| | *Kendall's $\tau'$* | | | |
| Static | 0.4340 | 0.5280 | 0.5705 | 0.4310 |
| Joint | 0.5075 | 0.6580 | 0.7187 | 0.7492 |
| EM-L | **0.9195** | **0.7880** | *0.8700* | *0.8564* |
| R-L | *0.9090* | *0.7160* | **0.8889** | **0.8903** |

Table 4: Comparison results with full supervision of the original datasets. Standard deviations are omitted to save space.

model with two backbones, indicating that retrieving examples from the training set could enhance the generalization, even with a narrow search scope. Secondly, the joint learning-based retrieval, EM-L, and R-L perform better than the static retrieval, which is even less effective than the vanilla model. We hold that this is because static retrieval fetches some examples with high semantic similarities but is detrimental to the downstream tasks. In contrast, the learnable retrieval methods, i.e. joint learning-based retrieval, EM-L, and R-L, are more likely to align with the goals of specific tasks.

*EM-L and R-L approaches train the retriever more effectively than static retrieval and joint learning-based retrieval.* At first, our proposed EM-L and R-L achieve significantly higher accuracy across different backbones, proving their effectiveness in fetching helpful examples and adapting to specific downstream tasks. Furthermore, on average, R-L outperforms EM-L, potentially due to its utilization of a more direct ranking loss that provides more significant signals and flexible guidance to the example retriever. Finally, it is worth noting that EM-L and R-L show smaller standard deviations on most datasets than other methods, we conjecture that the proposed training objectives enhance the stability of generalization by incorporating retrieval memory alongside parameters.

*The advantages of EM-L and R-L are more pronounced on challenging tasks,* such as sentence pair classification, and aspect-based sentiment analysis. In this regard, EM-L and R-L achieve improvements of more than 0.3 on most datasets for sentence pair classification and ABSA, whereas the improvement on the single-sentence classification ranges from 0.1 to 0.2, which gain further highlights the effectiveness of EM-L and R-L.

### 4.3 Consistency Experiments

The *Kendall's $\tau'$* defined in Eq. (11) on selected datasets are reported in Table 2, which measures the consistency between retrieval metrics of fetched examples and their auxiliaries to downstream tasks. Combing the results in Table 1, higher $\tau'$ of EM-L

and R-L indicates that they could prioritize more helpful examples according to their corresponding metrics and improve the performance by training more effective retrievers. However, retrieving examples according to static metrics and joint learning-based metrics may result in the inclusion of harmful examples in the final performance.

### 4.4 Auxiliary Experiment

We further conduct additional experiments in both 8-shot and full supervision settings to investigate the advantages of EM-L and R-L on different data scales. The results are presented in Table 3 and Table 4, respectively. It is obvious that EM-L and R-L consistently exhibit excellence in both settings. Particularly, we note a more significant improvement of our methods in the 8-*shot* setting, which manifests that the proposed training methods train the retriever more effectively, especially when the training data is scarce.

Moreover, another interesting phenomenon emerged: although EM-L and R-L achieve higher *Kendall's $\tau'$* in the full supervision setting, their improvements in text classification are comparatively

smaller compared to that in few-shot scenarios. We believe this can be attributed to the fact that the classifier in the full supervision setting is already well-trained so the potential improvement from a better retrieval memory is relatively limited.

## 5  Analysis

### 5.1  Effects of the Number of Retrieved Examples

To examine the effects of the number $m$ on various retriever training methods, we present line charts in Fig. 1 that depict the relationship between *Accuracy* and $m$. First, all the charts demonstrate retrieving examples could enhance the performance of few-shot text classification, except for a slightly lower accuracy of static retrieval and joint learning-based retrieval when $m$ takes specific values. This could be attributed to the instability of their training process. Second, most methods achieve their peak performance at $m = 5$ or $m = 10$. As $m$ continues to increase, the performance may start to deteriorate. We guess the reason is that retrieving too many examples increases the training difficulty. Third, we observe EM-L and R-L maintain sustaining advantages and stability as $m$ varies, which verifies their stronger supervision signals. Another observation is that the joint learning-based method falls behind the static method on LAP. This finding suggests that in certain tasks, a poorly trained learnable metric even exhibits inferior performance compared to a static metric.

### 5.2  Gradient Updates

In order to assess the supervision signals exerted on the retrievers by different methods, we quantify the average gradients of all retrievers' parameters. This measurement allows us to evaluate the guidance provided by each method to the retriever during the training process. Fig. 2 illustrates the percentage of training steps where the average gradients of all retrievers' parameters exceed the threshold of $1e - 6$. For clarity, we exclude static retrieval from this figure since its retriever has no trainable parameters[1]. Our analysis revealed that on certain datasets, the gradient norm of the joint learning-based retriever exceeds the threshold of $1e - 6$ for only about $40\%$ of the steps, whereas EM-L and R-L surpass this threshold in over $60\%$ of the steps. This observation suggests that both static and joint learning-

---

[1]This corresponds to a constant proportion of zero for steps with a gradient norm exceeding 1e-6.

based retrieval provide weaker supervision signals to the retrievers and suffer from severe vanishing issues in few-shot text classification while EM-L and R-L alleviate such limitations.

### 5.3  Case Study

Finally, we present an illustrative example from the LAP dataset along with the retrieved examples using different methods in Fig. 3. In the input sentence, the aspect term "*startup times*" is negative. Although static retrieval fetches a semantic similar example, it includes information that could potentially mislead the sentiment prediction, such as the term "*spectacular*". The joint learning-based retrieval retrieves an example that seems unrelated to the input sentence, possibly indicating that weak supervision signals for the retriever are prone to worse retrieval results. In contrast, our EM-L and R-L methods are capable of retrieving examples that may not possess high semantic similarity but are more beneficial for sentiment prediction.

## 6  Related Work

### 6.1  Retrieval-augmented Methods

Retrieval-augmented methods enhance the ability of the Pre-trained Language Models in processing various natural language tasks by fetching relevant examples from the training set or external knowledge base and prepending them with the original input. These methods have improved the performance of a lot of tasks, such as neural machine translation (Zhang et al., 2018; Cai et al., 2021; Li et al., 2022; Wang et al., 2022), question answering (Li et al., 2020; Karpukhin et al., 2020; Singh et al., 2021; Wang et al., 2022; Siriwardhana et al., 2023; Li et al., 2023; Hofstätter et al., 2023), dialog generation (Fan et al., 2021; Thulke et al., 2021; King and Flanigan, 2023), text classification (Izacard et al., 2022; Lewis et al., 2020), keyphrase generation (Gao et al., 2022), etc. According to retrieval metrics, these methods could be categorized as static retrieval methods and joint learning-based methods, which use a fixed retrieval metric and jointly learnable metric respectively.

Different from the above methods, which fetch relevant examples from the large-scale corpus, we propose two novel training objectives to retrieve examples in a restricted retrieval space and analyze their advantages. Following Singh et al. (2021); Izacard et al. (2022), we formulate the retrieval-augmented methods into a retriever and a classifier

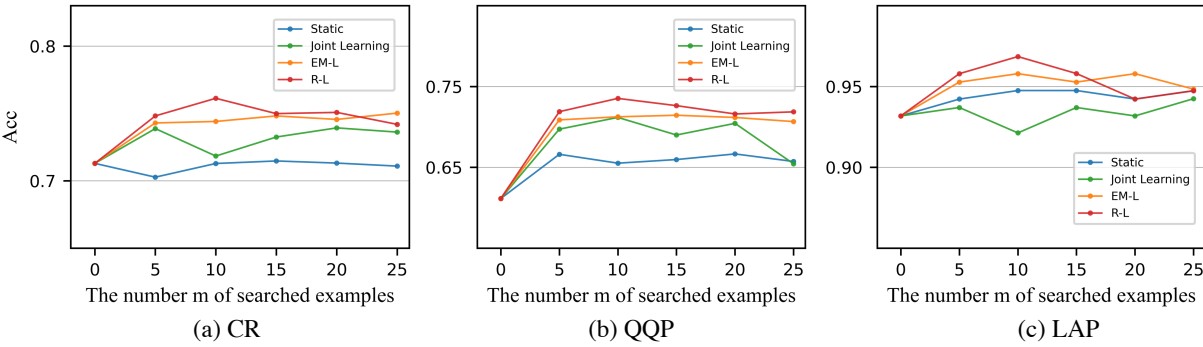

(a) CR

(b) QQP

(c) LAP

Figure 1: Effects of the number $m$ of retrieved examples. The results are average *Accuracy* on the validation set.

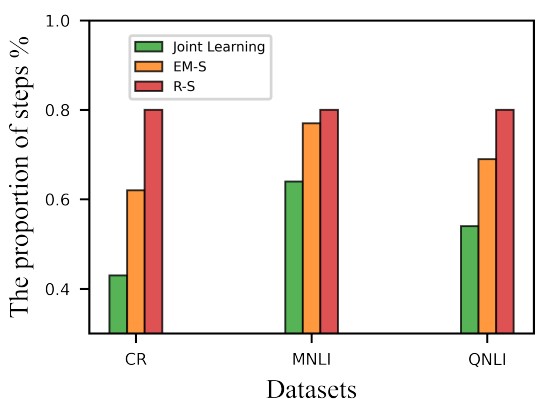

Figure 2: The proportion of steps in which the average gradient of retriever's all parameters is more than $1e-6$.

in Eq. (1) for a fair comparison.

## 6.2 Prompt Engineering

Fueled by the birth of large-scale language models (Brown et al., 2020), prompt-based learning (Liu et al., 2023) for the Pre-trained Language Models has been developed to convert different downstream tasks into cloze-style mask language model objectives, achieving impressive performance in text classification (Wang et al., 2021; Gao et al., 2021; Hambardzumyan et al., 2021; Lester et al., 2021; Schick et al., 2020; Schick and Schütze, 2021), sentiment classification (Seoh et al., 2021; Yan et al., 2021; Chen and Qian, 2020; Zhang et al., 2021), named entity recognition (Cui et al., 2021), relation extraction (Chen et al., 2022b,b), question answering (Lewis et al., 2019; Khashabi et al., 2020), commonsense reasoning (Shwartz et al., 2020), etc. Orthogonal to these studies of prompt learning, our paper focuses on the comparison of different retrieval methods, where prompt learning is just employed as a backbone.

## 6.3 Few-shot Text Classification

Few-shot Text Classification trains a classifier with limited data for each class, which can also predict unseen classes. Existing studies for few-shot text classification encompass various approaches such as prototypical networks (Jake et al., 2017), XLNet-based methods (Zhilin et al., 2019), (Ro)BERT(a)-based methods (Chen et al., 2020, 2022a), Pattern-exploiting training (Schick and Schütze, 2021), prompt tuning (Lester et al., 2021; Gao et al., 2021), etc. And common sub-tasks in text classification consist of intention classification, topic classification, sentiment classification, etc. We evaluate our methods on different text classification tasks, with a focus on adapting the idea of retrieval-augmented methods to the few-shot scenarios through the design of new training objectives.

## 7 Conclusion

This paper studies the retrieval-augmented methods for few-shot text classification and demonstrates the challenges which hinder their success: it is impossible to retrieve semantically similar examples by using an off-the-shelf metric and it is difficult to optimize a plausible metric by minimizing the standard cross-entropy loss. Accordingly, it proposes two novel training objectives, EM-L and R-L, which provide stronger supervision signals to train the retrieval metric effectively in few-shot scenarios. It is worth mentioning that the idea of searching within limited examples bears similarity to the concept of demonstration selection in recent large language models (LLMs). Exploring the application of our methods in LLMs holds promise for future research.

| Input: *Startup times* are incredibly long : over two minutes. The sentiment polarity of *startup times* was \<mask\> . |
|---|

| Methods | Predictions | | Retrieved Examples | Labels for Retrieved Examples |
|---|---|---|---|---|
| **Static** | positive | ✗ | The *internet speed* is spectacular. The sentiment polarity of *internet speed* was \<mask\> . | positive |
| **Joint** | positive | ✗ | That included the extra Sony Sonic Stage software , the speakers and the subwoofer I got -LRB- that WAS worth the money -RRB- , the bluetooth mouse for my supposedly bluetooth enabled computer , the extended life battery and the *docking port*. The sentiment polarity of *docking port* was \<mask\> . | neutral |
| **EM-L** | negative ✓ | | Its not just slow on the *internet*, its slow in general. The sentiment polarity of *internet* was \<mask\> . | negative |
| **R-L** | negative ✓ | | Another thing is that after only a month the *keyboard* broke and it costed $175 to send it in to fix it . The sentiment polarity of *keyboard* was \<mask\> . | negative |

Figure 3: Case Study. "Input" denotes an input sentence from LAP, "Predictions" represents the predicted sentiment polarities of different methods, and "Retrieved Examples" is the fetched examples with the highest metric in the training set. "Labels for Retrieved Example" denotes sentiment labels of the fetched examples.

## Limitations

There are three primary limitations of our methods. Firstly, EM-L and R-L require additional training time compared to existing retrieval methods. It is due to the alternation between the E-step and M-step in EM-L and the optimization of an additional loss of R-L. Specifically, the training time for EM-L per epoch is approximately 1.5 times that of static retrieval and 1.2 times that of joint learning-based retrieval. Similarly, the training time for R-L per epoch is about 1.8 times that of static retrieval and 1.5 times that of joint learning-based retrieval. Although our proposed methods require more time, they still fall within the acceptable range. Secondly, we didn't focus on designing more sophisticated templates for prompt engineering, as our main emphasis was on exploring different retrieval methods. Thirdly, we evaluate our methods in few-shot settings constructed from widely used datasets, rather than real-world scenes. This could limit the generalizability of our findings to practical applications.

## Acknowledgements

The research work is supported by the National Key R&D Plan No. 2022YFC3303303, the National Natural Science Foundation of China under Grant (No.61976204). This study is also supported by grants from the Major Key Project of PCL (Grant Number: PCL2022D01) and the CAAI Huawei MindSpore Open Fund. Xiang Ao is also supported by the Project of Youth Innovation Promotion Association CAS, Beijing Nova Program Z201100006820062.

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

# A Proof of the EM-L Method

**Proposition.** Optimizing the following two likelihood functions is equivalent in EM-L:

$$\max_{\theta,\phi} \prod_i^n P_{\theta,\phi}(y|\mathbf{x}_i) \iff$$
$$\max_{\theta,\phi} \sum_i^n \sum_j^m P_{\theta,\phi}(\mathbf{z}_j|\mathbf{x}_i, y) \log P_\theta(y|\mathbf{x}_i, \mathbf{z}_j),$$
$$\text{where } P_{\theta,\phi}(\mathbf{z}_j|\mathbf{x}_i, y)$$
$$:= \frac{P_\theta(y|\mathbf{x}_i, \mathbf{z}_j) P_\phi(\mathbf{z}_j|\mathbf{x}_i)}{\sum_j^m P_\theta(y|\mathbf{x}_i, \mathbf{z}_j) P_\phi(\mathbf{z}_j|\mathbf{x}_i)},$$

(12)

where $\mathbf{x}_i$ is the representation of the $i$-th sentence. For each $\mathbf{x}_i$, the retriever fetches $m$ examples from the corpus to assist $\mathbf{x}_i$ in text classification, where each example is represented as $\mathbf{z}_j$.

**Proof.** We first use variational inference to derive the lower bound of the original likelihood:

$$\max \prod_i^n P_{\theta,\phi}(y|\mathbf{x}_i)$$
$$\iff \max \log \prod_i^n P_{\theta,\phi}(y|\mathbf{x}_i)$$
$$= \max \sum_i^n \log P_{\theta,\phi}(y|\mathbf{x}_i)$$
$$= \max \sum_i^n \log \sum_j^m P_\theta(y|\mathbf{x}_i, \mathbf{z}_j) P_\phi(\mathbf{z}_j|\mathbf{x}_i)$$
$$= \max \sum_i^n \log \sum_j^m P_{\phi,\theta}(y, \mathbf{z}_j|\mathbf{x}_i)$$

(13)

Let $Q(\mathbf{z}_j)$ be a random distribution of $\mathbf{z}_j$:

$$\max \sum_i^n \log \sum_j^m P_{\phi,\theta}(y, \mathbf{z}_j|\mathbf{x}_i)$$
$$= \max \sum_i^n \log \sum_j^m Q(\mathbf{z}_j) \frac{P_{\phi,\theta}(y, \mathbf{z}_j|\mathbf{x}_i)}{Q(\mathbf{z}_j)} \quad (14)$$
$$\geq \max \sum_i^n \sum_j^m Q(\mathbf{z}_j) \log \frac{P_{\phi,\theta}(y, \mathbf{z}_j|\mathbf{x}_i)}{Q(\mathbf{z}_j)}$$

The last step is according to Jansen inequality and equals if and only if $Q(\mathbf{z}_j)$ is proportional to $P_{\theta,\phi}(y, \mathbf{z}_j|\mathbf{x}_i)$ and $c$ is a constant. Such a proportional relationship can be expressed as:

$$\frac{Q(\mathbf{z}_j)}{P_{\theta,\phi}(y, \mathbf{z}_j|\mathbf{x}_i)} = c, c \text{ is a constant}$$
$$\Longleftrightarrow cP_{\theta,\phi}(y, \mathbf{z}_j|\mathbf{x}_i) = Q(\mathbf{z}_j), \quad \forall i, j \tag{15}$$

Since $\sum_j Q(\mathbf{z}_j) = 1$, we can sum $\mathbf{z}$ on both sides of the equation:

$$\Longleftrightarrow c\sum_j^m P_{\theta,\phi}(y, \mathbf{z}_j|\mathbf{x}_i) = 1$$
$$\Longleftrightarrow c = \frac{1}{\sum_j^m P_{\theta,\phi}(y, \mathbf{z}_j|\mathbf{x}_i)} \tag{16}$$

Now we can derive a lower bound of $\prod_i^n P_{\theta,\phi}(y|\mathbf{x}_i)$ by substituting $c$ into Eq.(15) and then substituting $Q(\mathbf{z}_j)$ to Eq.(14):

$$Q(\mathbf{z}_j) = \frac{P_{\theta,\phi}(y, \mathbf{z}_j|\mathbf{x}_i)}{\sum_j^m P_{\theta,\phi}(y, \mathbf{z}_j|\mathbf{x}_i)}$$
$$= \frac{P_\theta(y|\mathbf{x}_i, \mathbf{z}_j)P_\phi(\mathbf{z}_j|\mathbf{x}_i)}{\sum_j^m P_\theta(y|\mathbf{x}_i, \mathbf{z}_j)P_\phi(\mathbf{z}_j|\mathbf{x}_i)} \tag{17}$$
$$= P_{\theta,\phi}(\mathbf{z}_j|\mathbf{x}_i, y)$$

$$\max \prod_i^n P_{\theta,\phi}(y|\mathbf{x}_i) \Longleftrightarrow$$
$$\max(\sum_i^n \sum_j^m Q(\mathbf{z}_j) \log P_{\theta,\phi}(y, \mathbf{z}_j|\mathbf{x}_i) \tag{18}$$
$$- \sum_i^n \sum_j^m Q(\mathbf{z}_j) \log Q(\mathbf{z}_j))$$

Since $P_{\theta,\phi}(y, \mathbf{z}_j|\mathbf{x}_i) = P_\theta(y|\mathbf{x}_i, \mathbf{z}_j)P_\phi(\mathbf{z}_j|\mathbf{x}_i)$, we can further simplify Eq.(18) as follows:

$$\max(\sum_i^n \sum_j^m Q(\mathbf{z}_j) \log P_{\theta,\phi}(y, \mathbf{z}_j|\mathbf{x}_i)$$
$$- \sum_i^n \sum_j^m Q(\mathbf{z}_j) \log Q(\mathbf{z}_j))$$
$$= \max(\sum_i^n \sum_j^m Q(\mathbf{z}_j) \log P_\theta(y|\mathbf{x}_i, \mathbf{z}_j)P_\phi(\mathbf{z}_j|\mathbf{x}_i)$$
$$- \sum_i^n \sum_j^m Q(\mathbf{z}_j) \log Q(\mathbf{z}_j))$$
$$= \max(\sum_i^n \sum_j^m Q(\mathbf{z}_j) \log P_\theta(y|\mathbf{x}_i, \mathbf{z}_j)$$
$$+ \sum_i^n \sum_j^m Q(\mathbf{z}_j) \log P_\phi(\mathbf{z}_j|\mathbf{x}_i)$$
$$- \sum_i^n \sum_j^m Q(\mathbf{z}_j) \log Q(\mathbf{z}_j))*$$
$$= \max(\sum_i^n \sum_j^m Q(\mathbf{z}_j) \log P_\theta(y|\mathbf{x}_i, \mathbf{z}_j))$$
$$= \max(\sum_i^n \sum_j^m P_\phi(\mathbf{z}_j|\mathbf{x}_i, y) \log P_\theta(y|\mathbf{x}_i, \mathbf{z}_j))$$
$$\tag{19}$$

Specifically, in the step denoted with $*$, $\sum_i^n \sum_j^m Q(\mathbf{z}_j) \log P_\phi(\mathbf{z}_j|\mathbf{x}_i)$ and $\sum_i^n \sum_j^m Q(\mathbf{z}_j) \log Q(\mathbf{z}_j))$ can be canceled out, because $Q(\mathbf{z}_j) = P_\phi(\mathbf{z}_j|\mathbf{x}_i, y) \approx P_\phi(\mathbf{z}_j|\mathbf{x}_i)$ in Eq. (17).

Further proof for convergence and equality of the original two optimizations is ordinary to derive as the proof of the EM algorithm, which is omitted here.

## B  Dataset Detail

### B.1  Original Datasets

All the retrieval methods are evaluated on three types of datasets: single-sentence classification, sentence pair classification, and aspect-based sentiment analysis (ABSA). The single-sentence classification consists of SST-2 (Socher et al., 2013), MR (Pang and Lee, 2004), CR (Hu and Liu, 2004), and TREC (Voorhees and Tice, 2000). The sentence pair classification includes QQP [2],

---
[2]https://quoradata.quora.com

| Dataset | Input | Output | Train | Test | Type |
|---------|-------|--------|-------|------|------|
| SST-2 | sentence $x$ | 1: positive
0: negative | 6,920 | 872 | sentiment classification |
| MR | sentence $x$ | 1: positive
0: negative | 8,662 | 2,000 | sentiment classification |
| CR | sentence $x$ | 1: positive
0: negative | 1,775 | 2,000 | sentiment classification |
| TREC | sentence $x$ | 0: Personality
1: Advisor
2: Conclusion
3: Human
4 :Assignment
5: Minute | 5,452 | 500 | question classification |
| QQP | sentence $x_1, x_2$ | 1: entailment
0: not entailment | 363,846 | 40,431 | paraphrase |
| QNLI | sentence $x_1, x_2$ | 1: entailment
0: not entailment | 104,743 | 5,463 | Natural Language Inference |
| MNLI | sentence $x_1, x_2$ | 2: entailment
1: neutral
0: contradiction | 392,702 | 9,815 | Natural Language Inference |
| SNLI | sentence $x_1, x_2$ | 2: entailment
1: neutral
0: contradiction | 549,367 | 9,842 | Natural Language Inference |
| RES | sentence $x$, aspect a | 2: positive
1: neutral
0: negative | 3,044 | 800 | aspect-based sentiment analysis |
| LAP | sentence $x$, aspect a | 2: positive
1: neutral
0: negative | 3,048 | 800 | aspect-based sentiment analysis |

Table 5: Dataset details. The column labeled "Train" represents the number of instances in the original training set, while "Test" denotes the number of instances in the test set. The "Type" column describes the task type associated with each dataset.

QNLI (Rajpurkar et al., 2016), SNLI (Bowman et al., 2015), and MNLI (Williams et al., 2017). The aspect-based sentiment analysis datasets are RES (Manandhar, 2014) and LAP (Manandhar, 2014). Particularly, for SST-2, MNLI, and QNLI from GLUE (Wang et al., 2018) and SNLI, we utilize their original validation sets for testing purposes.

## B.2 Few-shot Datasets

Following the few-shot setting of Gao et al. (2021), we randomly select 16 or 8 examples from the training set to create 16-shot or 8-shot experiments. Specifically, we generate five distinct few-shot datasets using different seeds and train models on each of them. It is noted that we use consistent five seeds on different datasets and retrieval methods to conduct a fair comparison. The best model is chosen based on the validation results, and the av-

erage evaluation scores on the original test set are reported.

## C Experimental Settings

### C.1 Hyper-parameter Selection

We adopt grid search to choose the hyper-parameters of different methods. Specifically, the learning rates are taken from $\{1e-5, 2e-5, 5e-5\}$, the batch sizes are from $\{4, 8, 16\}$, and the numbers of retrieved examples are taken from $\{5, 10, 15\}$. The parameter $t$ that determines the update frequency of loss $\mathcal{L}_R$ is searched from $\{5, 10, 15\}$. The loss coefficient $\lambda$ in ranking-based loss is set to $\{0.5, 1, 2\}$. For each dataset, we set the max training steps as 800 steps and use early stopping to avoid over-fitting. In each trial, we validate the model in each epoch and save the best checkpoint.

We adopt the AdamW optimizer and accumulate gradients for each batch. The code is imple-

| Dataset | Template | Label |
|---------|----------|-------|
| SST-2 MR CR | Input sentence $x$, it was \<mask\>. | 1: positive → good 
 0: negative → terrible |
| TREC | Input sentence $x$, it was \<mask\>. | 0: Personality→Personality 
 1: Advisor→Advisor 
 2: Conclusion→Conclusion 
 3: Human→Hum 
 4 :Assignment→Assignment 
 5: Minute→Minute |
| QQP QNLI | Input sentence $x_1$, \<mask\>, $x_2$. | 1: entailment → Yes 
 0: not entailment → No |
| MNLI SNLI | Input sentence $x_1$, \<mask\>, $x_2$. | 2: positive→ positive 
 1: neutral → neutral 
 0: negative → nagative |
| RES LAP | Input sentence $x$, the a was \<mask\>. | 2: positive→ positive 
 1: neutral → neutral 
 0: negative → nagative |

Table 6: Templates and label words for different datasets that we used for prompt-based fine-tuning.

mented with PyTorch 1.9.0 and transformers 4.1.1 and launched on an Ubuntu server with a single NVIDIA Tesla V100 (32G) or NVIDIA 4090. In addition, we will test our model with Mindspore, which is a new deep-learning framework[3].

## C.2 Templates of Prompt-based Fine-tuning

We use RoBERTa-Large (Liu et al., 2019)[4] with 1024 dimensions to encode the input sentences with the related template. The templates for various datasets are shown in Table 6. Since our main aim is to investigate the difference among retrieval methods, we adopt the widely used and effective templates for these tasks in prompt-based fine-tuning refer to Gao et al. (2021). The specific templates are shown in Table 6.

---

[3]https://www.mindspore.cn/
[4]https://github.com/huggingface/transformers