# OpenReview forum: "Retrieval-Augmented Few-shot Text Classification"
_EMNLP/2023/Conference — EMNLP 2023 Findings_

### Official Review · Reviewer_VkCo · 2023-07-29

**Soundness:** 4

**Excitement:**

4: Strong: This paper deepens the understanding of some phenomenon or lowers the barriers to an existing research direction.

**Paper Topic And Main Contributions:**

The paper studies the retrieval-augmented few-shot classification. Motivated by the fact that the existing retrieval methods all suffer from data scarcity and vanishing gradient problem, the paper proposed an EM algorithm which alternatively retrieves the best m training data and update the parameters by maximizing the log-likelihood. Besides, it also proposed a R-L algorithm which minimizes the discrepancy between the retriever and the classifier. The comprehensive empirical studies verifies the effectiveness of the proposed two techniques.

**Questions For The Authors:**

1. It is not clear why EM-L method does not suffer from the data scarcity problem.
2. Is it possible to combine EM-L and R-L? If yes, why the paper does not combine them? If not, is there any reason for that?
3. Will the EM-L suffer from worse training efficiency? Since it alternatively performs the E-step and M-step.

**Reasons To Accept:**

1. The paper studies a very interesting problem, namely retrieval-based few-shot text classification, which is of great significance to the NLP researchers and practitioners.
2. The paper is technically sound and the proposed algorithm effectively mitigate the shortcomings of the existing algorithms.
3. The paper is well written and easy to follow.
4. The experiments are very solid and comprehensive. The results look great.

**Reasons To Reject:**

1. There are still some unclear issues to be better elaborated (e.g., why the paper does not combine EM-L and R-L?).

**Reproducibility:**

5: Could easily reproduce the results.

**Reviewer Confidence:**

3: Pretty sure, but there's a chance I missed something. Although I have a good feel for this area in general, I did not carefully check the paper's details, e.g., the math, experimental design, or novelty.

---

> ### Author Rebuttal · Authors · 2023-08-29
>
> Thank you for your valuable comments. We will improve this paper according to your suggestions. We answer your questions in order as follows.
>
> **W1: The combination of EM-L and R-L**
>
> First, we didn't combine EM-L and R-L because of their different calculation methods of relevance scores and different optimization objectives.  Although the idea of EM-L and R-L are consistent, the relevance scores of EM-L are computed by estimating the posterior distribution of the latent variable (in Eq. (6)), while that of R-L are calculated by the examples retriever directly.
>
> Second, we didn't combine EM-L and R-L because they consider the retrieval augmented classification tasks from different perspectives. In the EM-L approach, the retrieved examples are treated as latent variables (lines 282 to 285). In the R-L approach, the retrieved examples are treated as a ranking process (lines 312 to 313).
>
> Of course, according to the reviewer's suggestion, we are happy to conduct this additional experiment: we combine EM-L and R-L to obtain the "EM-L+R-L" model and report the comparison results on selected datasets in Table D. As the results show, "EM-L+R-L" has no obvious improvement compared with EM-L and E-L.
>
> We will add the above reasons in the final version for a more comprehensive experimental analysis.
>
> **Table D**. Comparison results. "EM-L + R-L" denotes the combination of EM-L and R-L.
> | Model      | TREC      | CR        | SST-2     | Res14     | Lap14     |
> | ---------- | --------- | --------- | --------- | --------- | --------- |
> | EM-L       | 92.13     | **90.00** | 91.31     | 73.74     | 76.02     |
> | R-L        | **92.86** | 89.93     | **91.58** | **76.79** | 75.59     |
> | EM-L+R-L | 92.54     | 89.88     | 91.24     | 73.95     | **76.45** |
>
> **Q1: Data scarcity problem of EM-L**
>
> As shown in Eq. (6),  on the one hand, EM-L could compute the conditional probabilities of the latent variable based on m retrieved examples, which are more comprehensive. On the other hand, EM-L takes into consideration the auxiliary function $P_θ(y|x, z_j)$ of retrieved example $z_j$ to the downstream task. The above two characteristics provide more evidence for training than existing retrieval methods and alleviate the data scarcity problem, so as to better train the retriever and select more helpful examples.
>
> **Q2: Why not combine EM-L and R-L**
>
> Please refer to the answer to W1.
>
> **Q3: Training efficiency**
>
> Due to the limited number of retrieved examples and few-shot training data, both the posterior distribution estimation in E-step and the maximization in M-step will not increase too much time. The extra training time caused by alternatives still falls within the acceptable range. Please refer to the limitation section and answer for **W3 to Reviewer U8Tr** for specific efficiency comparison.

---

### Official Review · Reviewer_U8Tr · 2023-07-31

**Soundness:** 4

**Excitement:**

4: Strong: This paper deepens the understanding of some phenomenon or lowers the barriers to an existing research direction.

**Paper Topic And Main Contributions:**

This work utilized the idea of EM algorithm to model and to jointly train the retrieval-augmented methods for a better classifier using a retriever.
The latent distribution in Eq. 6 can be modeled and estimated by the classifier and retriever in the E-step, subsequently, both the classifier and retriever are jointly optimized in the M-step from the variational distribution (evidence lowerbond). In addition, the paper also proposed a Ranking-based Loss where a high probability prediction given the augmented retrieval results is supposed to have a higher probability to generate these retrieval results.

The main contribution of this paper is introducing the idea of EM algorithm to model the retrieval-augmented methods and being able to show this is a data-efficient way to optimize both classifier and retriever than previous methods such as Joint Learning based Retrieval. Besides, this work shows the modeling can alleviate the vanishing gradient issue in training deep models, presumably by offering more training signals (with the number of loss terms proportionally to the sampled retrieved results m, I assume).

**Questions For The Authors:**

Other than using the Jansen inequality to show lower bound, would the maximization objective be obtained from normal EM derivation (using ELBO)? What are the differences between your M objective with standard EM (I cannot tell by looking at the final form)?

**Reasons To Accept:**

The work is theoretically sound and could be significant for few-shot retrieval-augmented classification, potentially also generalizable to retrieval-augmented LLM tasks as well (as the author also indicated in the conclusion). The results are promising in data scarcity settings (especially good in 8-shot experiments) for a limited number of retrieval sample (m as small as 10 is well enough, according to Figure 1).

**Reasons To Reject:**

* The formula notations are not clean. E.g., what is "s" in x^s at line 140? Can we write y in Y=y or y_i? And y should be explained when it first occurs in Eq. 1. Also, many unnecessary spaces in Appendix formulas, using math symbols instead of \text for in-formula texts, and not using \left( and \right) makes the derivation unpleasant to read. Plus, there is a "*" (star) at the final steps of Eq. 19 for no reason. Overall, these issues are minor and non-critical, but they indicate the potential for polishes.

* The retrieved example in R-L case of Figure 3 looks dubious, as it is irrelevant to the input.

* The training time is important aspects in the evaluation of proposed methods. I suggest the authors give full results on training times instead of mentioning it in the Limitations section (including the effects of m and training samples n).

* Lacking implementation details in terms of f_retr and f_cls functions, are they separate RoBERTa? Or one is stacked on top of the other? Also, I would recommend the authors to state the willingness of publishing their code, as they are critical for understanding the details of this work.

**Reproducibility:**

4: Could mostly reproduce the results, but there may be some variation because of sample variance or minor variations in their interpretation of the protocol or method.

**Reviewer Confidence:**

3: Pretty sure, but there's a chance I missed something. Although I have a good feel for this area in general, I did not carefully check the paper's details, e.g., the math, experimental design, or novelty.

---

> ### Author Rebuttal · Authors · 2023-08-29
>
> Thank you for your valuable comments.  We will solve all the formulation problems and clarify the details in the final version. We answer your questions in order as follows.
>
> **W1：Formulation notations**
>
> In line 140, $s$ in $x^s$ and $y^s$ denote an example sentence, while $x^s$ and $y^s$ respectively stand for the input and the label of the example s. We consider $y$ as a distribution and $y_i$ is a specific label. By your guidance, we will remove the unnecessary spaces in Appendix formulas and follow your suggestions to use math symbols. The employment of the "*" symbol in Eq. (9) serves to indicate a forthcoming description of this step, which is shown in lines 945-946. We will revise these problems in the final version.
>
> **W2: Retrieved examples**
>
> Our study found that some helpful retrieved examples do not necessarily maintain a very high semantic similarity with the input sentence. On the contrary, examples with very high semantic similarities may be harmful to downstream performance (refer to lines 389-394). EM-L and R-L tend to exchange a little semantic similarity for greater help to downstream tasks. Furthermore, in Figure 3, the similarity between the retrieved example by R-L and the input sentence lies in that both of them describe a certain high attribute (the "setup time" or "costs") thus expressing negative sentiment polarities.
>
> **W3: Training time**
>
> (1)In alignment with your insightful recommendations, we further report the training times on selected datasets in Table B. The comparison results meet the results acknowledged in the limitations section.
>
> (2)We also report the training times of different models when the number m of retrieved examples ranges from $1$ to $9$, which are shown in Table C.
>
> **Table B**. Training times under the backbone of prompt learning on selected datasets. The number m of retrieved examples is $5$.
> | Training Time (ms/epoch) | SST-2    | MR       | CR       | TREC      | QNLI     | Res14    | Lap14    | Avg.     |
> | ------------------------ | -------- | -------- | -------- | --------- | -------- | -------- | -------- | -------- |
> | Static                   | 317.7609 | 407.1764 | 419.9924 | 1155.6439 | 388.6265 | 498.2818 | 529.7172 | 531.0284 |
> | Joint                    | 416.3982 | 474.4482 | 385.0958 | 1127.0368 | 396.7046 | 584.0690 | 527.6005 | 558.7647 |
> | EM-L                     | 401.8793 | 626.1048 | 553.6871 | 1329.2260 | 471.6467 | 664.9293 | 639.2867 | 669.0948 |
> | R-L                      | 568.6276 | 739.5091 | 671.3948 | 1500.2369 | 559.8521 | 797.4810 | 731.7583 | 795.5514 |
>
>
> **Table C**. Training times under the backbone of prompt learning when m ranges from $1$ to $9$ on Lap14.
> | Training Time (ms/epoch) | m=1       | m=3      | m=5      | m=7       | m=9      |
> | ------------------------ | --------- | -------- | -------- | --------- | -------- |
> | Static                   | 517.0159  | 525.0875 | 529.7172 | 584.45692 | 572.6985 |
> | Joint                    | 517.5697  | 523.3082 | 527.6005 | 529.60372 | 572.5355 |
> | EM-L                     | 602.61846 | 636.1904 | 639.2867 | 679.8043  | 695.4593 |
> | R-L                      | 706.64191 | 730.2668 | 731.7583 | 739.98213 | 772.3598 |
>
> **W4: Details of $f_{retr}$ and $f_{cls}$**
>
> In our experiment, we configured $f_{retr}$ and $f_{cls}$ functions as separate RoBERTa. And we will consider your suggestion to release our code.
>
> **Q1: About proof**
>
> Both EM and ELBO are motivated by the idea of Variational Bayesian Inference. In our derivation process, we provide a detailed presentation of the variational inference procedure by employing Jansen’s inequation to emphasize the practical significance of Eq.(7). Thus obtaining the maximization objective using ELBO is fully valid and justifiable.
>
> Unlike the standard EM algorithm, our results explicitly incorporate Y to emphasize the forward inference process of X -> Z -> Y in our algorithm. This is also done for the convenience of readers to understand and reproduce our algorithmic procedure. After all, deriving the loss function based on the original EM algorithm is not as intuitive.

---

### Official Review · Reviewer_uG9t · 2023-08-02

**Soundness:** 3

**Excitement:**

4: Strong: This paper deepens the understanding of some phenomenon or lowers the barriers to an existing research direction.

**Missing References:**

(1) A large portion of the references come from arXiv.
(2) Some references are not cited in the text. For example Yifan Gao, Qingyu Yin, Zheng Li, Rui Meng, Tong Zhao, Bing Yin, Irwin King, and Michael Lyu. 2022. Retrieval-augmented multilingual keyphrase generation with retriever-generator iterative training. In Findings ofthe Association for Computational Linguistics: NAACL 2022, pages 1233–1246.

**Paper Topic And Main Contributions:**

This paper addresses the problem of weak supervision signals for the retriever and insufficient data. They propose two novel training objectives, namely Expectation Maximization-based Loss (EM-L) and Ranking-based Loss (R-L), for learning to retrieve examples from a limited space more effectively.The main contributions of this paper are as follows:
They discuss the weak supervision signals and gradient vanishing problem encountered by existing retrieval methods minimizing the standard cross-entropy loss,
They introduce two novel training objectives, namely EM-L and R-L, which optimize the retriever more effectively, thus recalling more productive examples from a limited space.
Extensive experiments and analyses demonstrate that the proposed methods achieve better performance on few-shot text classification and alleviate the supervision insufficiency and gradient vanishing issues.

**Questions For The Authors:**

(1) Some of the symbols in Equation 1 are not explained, please add them.
(2) Regarding the author's mentioned solution to weak supervision signals for the retriever and insufficient data, can it be simply interpreted as using only quality data and discarding the rest?
(3) This paper lacks an overall framework figure.

**Reasons To Accept:**

(1) The method proposed in this paper have some practical significance.
(2) The ideas and methods of the paper are clear.
(3) The experiments in this paper are sufficient and the experimental angles are varied.

**Reasons To Reject:**

(1) The overall organization of the article needs to be adjusted, e.g., “Revisiting Retrieval-augmented Methods in Few-shot Learning” and methods both take up one page and there is duplication between “Revisiting Retrieval-augmented Methods in Few-shot Learning” and related work.
(2) The dataset section does not explicitly give which dataset was used.The baseline section does not give a clear method of comparison and does not compare to similar studies.
(3) The results of the experimental part show a simple form.

**Reproducibility:**

3: Could reproduce the results with some difficulty. The settings of parameters are underspecified or subjectively determined; the training/evaluation data are not widely available.

**Reviewer Confidence:**

3: Pretty sure, but there's a chance I missed something. Although I have a good feel for this area in general, I did not carefully check the paper's details, e.g., the math, experimental design, or novelty.

**Typos Grammar Style And Presentation Improvements:**

(1) In model formulation of section 2.1, the sentence “where x and zj denote the representations of original input and a retrieved example from the training set, fclf and fretr serve as…” lacks a conjunction.
(2) In last paragraph of section 3.4, the sentence “where λ > 0 is a hyperparameter to trade off both loss terms, step denotes the training steps” lacks a conjunction.

---

> ### Author Rebuttal · Authors · 2023-08-29
>
> We appreciate your valuable suggestions, which undoubtedly contribute to enhancing the clarity and coherence of our paper. We structure our responses based on distinct categories of questions, including article organization, comparison with similar studies, missing references, etc.
>
> **1. Article organization and unclear details**
>
> **W1: We are willing to adjust the length of Section 2 and Section 3 and try to mitigate redundancy between Section 2 and Section 6.** Given our endeavor to claim the limitation of existing retrieval methods and the challenges they faced in few-shot scenarios in Section 2 and the advantages of EM-L and R-L in Section 3, the length has extended slightly. An overlap between Section 2 and Section 6 has inadvertently arisen because both of them delineate some representative retrieval-augmented methods but Section 6 is more comprehensive.
>
> **W2: We will improve the details of datasets and baseline methods in Section 4.1.**
>
> We have put details of datasets and the construction methodologies applied to create few-shot datasets in Appendix B due to space constraints. Specifically, in our main experiment, we evaluate different models on $10$ datasets, categorized as follows: single-sentence classification (**SST-2**, **MR**, **CR**, and **TREC**), sentence pair classification (**QQP**, **QNLI**, **MNLI**, and **SNLI**), and aspect-based sentiment classification (**Res14**, and **Lap14**).
>
> We primarily compare three baselines, namely **Vanilla**, **Static**, and **Joint**, to prove the effectiveness of EM-L and R-L. All the methods acquire sentence representations either via Pre-trained Language Models (PLMs) or prompt learning using manual templates. **Vanilla** is a simple model without retrieval, which only adopts a feed-forward neural network as a text classifier to identify the category of each sentence. **Static** consists of a text classifier and an examples retriever, the operational mechanics of which are explained in Eq. (1) and Eq. (3).  Similarly, **Joint** also consists of a text classifier and an examples retriever, which are explained in Eq. (1) and Eq. (4). All the three baselines are optimized by minimizing a cross-entropy loss of text classification, as described in Eq. (2).
>
> **W3: To alleviate the simple experimental form, we shall add the supplementary results and more 8-shot experiments** in the final version, if space permits. We will consider including these supplementary materials in the Appendix If space constraints arise.
>
> **Q1: We will add explanations for several symbols in Eq. (1), as outlined below**:  $\theta$ and $\phi$ denote the trainable parameters of the text classifier and examples retriever. $z_j$ is the $j$-th retrieved example. $f_{clf}$ and $f_{retr}$ serve as the text classifier and the example retriever, both comprising feed-forward neural networks. $y$ corresponds to the class associated with input $x$. The operation $\oplus$ signifies concatenation, and the term "softmax" refers to the normalized exponential function.
>
> **Q3: We are willing to supplement a figure about the model overview if necessary.**  Since we have concentrated on different retrieval processes under the same prevailing retrieval framework, we paid more attention to the theory statement.
>
> Your suggestions for article organization are helpful to us, and we will improve it and fix all the typos.
>
> **2. Comparison with other similar studies**
>
> **W2: Following your valuable recommendations in W2, we add two additional baselines.**
>
> (1) **Vanilla+KNN** combines a KNN mechanism with the Vanilla Baseline. It retrieves the top $m$ relevant examples and merges their labels with the logits derived from Vanilla according to the relevance scores.
>
> (2) **Gao et al. 2022 (NAACL) [1]** proposed a Retriever-Generator Iterative Training method for the keyphrase generation task, which mines parallel passage pairs with high qualities according to retrieval scores. This method could be directly helpful for training the keyphrase generation module and is executed based on Parallel data $D_{PAR}$ and Non-parallel data $D_{NP}$. However, the context delineated in Gao et al. 2022 (NAACL) diverges from our present investigation, resulting in the nonexistence of the aforesaid $D_{PAR}$ and $D_{NP}$ within our study.
>
> In response to the reviewers' mention of Gao et al. 2022 (NAACL) in the missing references, we construct Parallel data $D_{PAR}$ and Non-parallel data $D_{NP}$ for our task to create a new baseline. Then the iterative algorithm elucidated by Gao et al. 2022 (NAACL) is incorporated to facilitate the training of our retriever and text classifier.
> Specifically, for each training example $x$, we first use the pre-trained language model to compute the semantic similarities among $x$ and other examples in the training set. We then select examples whose similarities with $x$ exceed 0.5 to form $D_{PAR}$ while the remaining examples form $D_{NP}$. Finally, we adopt the iterative algorithm in Gao et al. 2022 (NAACL)  to train our model. We set the iterations as $5$ across different datasets.
>
> **We make an additional analysis compared with Vanilla+KNN and Gao et al. 2022 (NAACL).** According to the results reported in Table A, **Vanilla+KNN** performs better than Vanilla, which proves the advantages of retrieval. However, Vanilla+KNN is inferior to Vanilla on CR. We think that the KNN fetches examples with high semantic similarities but harms the performance of downstream tasks, which is similar to the Static baseline analyzed in lines 389 to 392.
> **Gao et al. 2022 (NAACL) [1]** performs better than Vanilla and shows comparable performance on most datasets. However, on most datasets, Gao et al. 2022 (NAACL) [1] is less effective than EM-L and R-L. The reason might be that there is not enough initial Parallel data $D_{PAR}$ in our few-shot setting, which limits the exertion of the iterative algorithm.
>
> **Table A**. The comparison results with the two new baselines on selected datasets.  The parameters of the two baselines are maintained consistent with other baselines. We adopt the backbone of prompt learning.
> | Model                       | TREC         | CR           | SST-2        | Res14        | Lap14        |
> | --------------------------- | ------------ | ------------ | ------------ | ------------ | ------------ |
> | Vanilla                 | 87.20        | 88.36        | 84.84        | 72.05        | 71.81        |
> | Static                  | 90.95        | 87.06        | 88.60        | 70.95        | 73.01        |
> | Joint                   | 90.57        | 86.76        | 90.71        | 71.07        | 73.32        |
> | EM-L                    | 92.13 | 90.00 | 91.31 | 73.74 | 76.02 |
> | R-L                     | **92.86**    | **89.93**    | **91.58**    | **76.79**    | **75.59**    |
> | **Vanilla+KNN**             | 90.38        | 85.79        | 91.24        | 73.35        | 72.95        |
> | **Gao et al. 2022 (NAACL)** | 91.42        | 84.79        | 87.09        | 72.28        | 75.11        |
>
> [1] Yifan Gao, Qingyu Yin, Zheng Li, Rui Meng, Tong Zhao, Bing Yin, Irwin King, and Michael Lyu. 2022. Retrieval-augmented multilingual keyphrase generation with retriever-generator iterative training. In Findings of the Association for Computational Linguistics: NAACL 2022, pages 1233–1246.
>
> **3. Solution to weak supervision signals and insufficient data.**
>
> **Q2**: Yes, the proposed two retrieval methods could retrieve examples with high qualities from a limited retrieval space, which can improve the performance of the downstream tasks more effectively. The rest of the examples after retrieval will remain unutilized for downstream tasks.
>
> **4. Missing references**
>
> Our study focuses on different retrieval-augmented methods for few-shot text classification, so some studies that only concentrate on either retrieval-augmented methods or few-shot text classification are not included in the references. Following the reviewer's suggestions, we will cite the following similar published papers and check our references carefully.
>
> [1] Yifan Gao, Qingyu Yin, Zheng Li, Rui Meng, Tong Zhao, Bing Yin, Irwin King, and Michael Lyu. 2022. Retrieval-augmented multilingual keyphrase generation with retriever-generator iterative training. In Findings of the Association for Computational Linguistics: NAACL 2022, pages 1233–1246.
>
> [2] Li S, Gao Y, Jiang H, et al. Graph Reasoning for Question Answering with Triplet Retrieval[C]// Findings of the Association for Computational Linguistics: ACL 2023, pages 3366--3375.
>
> [3] Hofstätter S, Chen J, Raman K, et al. Fid-light: Efficient and effective retrieval-augmented text generation[C]//Proceedings of the 46th International ACM SIGIR Conference on Research and Development in Information Retrieval. 2023: 1437-1447.
>
> [4] King B, Flanigan J. Diverse Retrieval-Augmented In-Context Learning for Dialogue State Tracking[C]//Findings of the Association for Computational Linguistics: ACL 2023. 2023: 5570-5585.

---

### Meta-Review · Area_Chair_whwr · 2023-09-17

**Recommendation:** 2

**Metareview:**

The reviewers saw both strengths and weaknesses in the submitted version of the paper. The authors have provided a rebuttal that appeared to alleviate some concerns. After some discussion, the overall verdict was leaning toward an acceptance of this submission. However, there are still some issues to be addressed as raised by Reviewer uG9t. So, I am leaning toward a Borderline Sound on this submission.

---

### Decision · Program_Chairs · 2023-10-07

**Decision:**

Accept-Findings

**Comment:**

The reviewers saw both strengths and weaknesses in the submitted version of the paper. The authors have provided a rebuttal that appeared to alleviate some concerns. After some discussion, the overall verdict was leaning toward an acceptance of this submission. However, there are still some issues to be addressed as raised by Reviewer uG9t. So, I am leaning toward a Borderline Sound on this submission.